# An Update on the Surveillance of Livestock Diseases and Antimicrobial Use in Sierra Leone in 2021—An Operational Research Study

**DOI:** 10.3390/ijerph19095294

**Published:** 2022-04-27

**Authors:** Fatmata Isatu Bangura (Turay), Amara Leno, Katrina Hann, Collins Timire, Divya Nair, Mohamed Alpha Bah, Sahr Raymond Gborie, Srinath Satyanarayana, Jeffrey Karl Edwards, Hayk Davtyan, Sorie Mohamed Kamara, Amadu Tejan Jalloh, David Sellu-Sallu, Joseph Sam Kanu, Raymonda Johnson, Noelina Nantima

**Affiliations:** 1Livestock and Veterinary Services Division, Ministry of Agriculture and Forestry, Youyi Building, Brookfields, Freetown 00232, Sierra Leone; lenoamara87@yahoo.fr (A.L.); medalphabah2014@gmail.com (M.A.B.); srgborie@gmail.com (S.R.G.); soriesl@yahoo.com (S.M.K.); jallohtejan770@yahoo.com (A.T.J.); davidsellusallu@gmail.com (D.S.-S.); 2Emergency Center for Transboundary Animal Diseases (ECTAD), Food and Agriculture Organization of the United Nations (FAO), Freetown 00232, Sierra Leone; noelinanantima@yahoo.com; 3Sustainable Health Systems, Freetown 00232, Sierra Leone; hann.katrina@gmail.com; 4International Union Against TB and Lung Disease (The Union), 75006 Paris, France; collins.timire@theunion.org (C.T.); divya.nair@theunion.org (D.N.); ssrinath@theunion.org (S.S.); 5Department of Global Health, The University of Washington, Seattle, WA 98195, USA; jeffreykedwards@gmail.com; 6Tuberculosis Research and Prevention Center, Yerevan 0014, Armenia; haykdav@gmail.com; 7National Disease Surveillance Program, Ministry of Health and Sanitation, Sierra Leone National Public Health Emergency Operations Centre, Cockerill, Wilkinson Road, Freetown 00232, Sierra Leone; samjokanu@yahoo.com; 8Ministry of Agriculture and Forestry, Youyi Building, Brookfields, Freetown 00232, Sierra Leone; raymonda.johnson.rj@gmail.com

**Keywords:** One Health, antimicrobial resistance, operational research, West Africa, surveillance, SORT-IT, Sierra Leone

## Abstract

In Sierra Leone, in 2020, a study by the Livestock and Veterinary Services Division (Ministry of Agriculture and Forestry) on the surveillance system of animal diseases and antimicrobial use found poor reporting. Of the expected weekly districts reports, <1% were received and only three of the 15 districts had submitted reports occasionally between 2016 and 2019. Following this, staff-capacity-building on reporting was undertaken. In 2021, we reassessed the improvement in reporting and used the reports to describe livestock diseases and antimicrobials utilized in their treatment. Between March and October 2021, 88% of expected weekly reports from all 15 districts were received. There were minor deficiencies in completeness and consistency in the terminology used for reporting animal disease and antimicrobials. Available reports showed that 25% of the livestock had an infectious disease, and a quarter of the sick animals had received an antimicrobial drug. Most animals received antimicrobials belonging to World Organization for Animal Health’s “veterinary critically important” category (77%) and World Health Organization’s “critically” (17%) and “highly important” (60%) categories for human health. These indicate a significant improvement in the animal health surveillance system and highlight the need for enhanced antimicrobial stewardship to prevent misuse of antimicrobials that are significant in animal and human health.

## 1. Introduction

The consumption of antimicrobials is increasing globally at an unprecedented pace [1]. Approximately two-thirds of the worldwide antimicrobial consumption is in the animal industry [2]. Directly related to the higher consumption of antimicrobials is a corresponding rise in the prevalence of antimicrobial resistance (AMR), which adversely impacts the economy, human health, food security, and related social–economic factors [1,3]. Resistance to anthelmintics, which has been observed in animals for more than four decades, is now being reported among humans [4,5,6].

Universally, human diets are progressively becoming animal-protein-based, increasing the demand for animal products and escalating the use of antimicrobials in animals, particularly livestock [7,8]. When used judiciously for preventing and treating illness, antimicrobials lower production expenses and improve profits [9]. However, misuse of antimicrobials can lead to the expansion of AMR [10].

The World Health Organization (WHO), Food and Agriculture Organization of the United Nations (FAO), and the World Organization for Animal Health (also known as Office International des Epizooties, OIE) collectively have recognized the escalating danger of AMR originating from the livestock industry. In 2015, these organizations collaboratively developed the Global Action Plan on AMR with five strategic objectives. One of the primary objectives is to focus on the “optimized” use of antimicrobials in animal health practices. Key to these goals is gathering valuable data for operational and strategic decision-making that can be used going forward [11]. The OIE strategy on AMR with the prudent use of antimicrobials and the FAO action plan on AMR both emphasize the importance of surveillance systems, monitoring, and record-keeping related to antimicrobial use in animals in combating AMR [12,13]. In addition, global efforts for tackling AMR in animals gained traction in 2019, culminating with the publication of guidance for responsible and prudent use of anthelmintic chemicals to help control anthelmintic resistance in grazing livestock species by the OIE in 2021 [14].

Despite these global recommendations, developing countries have suboptimal surveillance systems with data deficiencies on antimicrobial use among animals [15,16,17]. In particular, there is a lack of data on the use of antimicrobials belonging to the OIE antimicrobial categories of “importance” and WHO antimicrobial categories of “concern” from developing countries [18,19]. In addition, there is minimal information regarding antimicrobial use, anthelmintic use, and the prevalence of AMR or anthelmintic resistance (AR) from West Africa [20].

In response to concerns regarding optimal use of antimicrobials and AMR, the Sierra Leone Ministry of Health and Sanitation (MOHS), Ministry of Agriculture and Forestry (MAF), and the Environmental Protection Agency (EPA) published the national strategic plan for combating AMR in 2018. The goals of this national plan have been adapted from the Global Action Plan on AMR to suit the country context. They include improving oversight, surveillance, and education for optimizing antimicrobial use in the country [21].

Currently, in Sierra Leone, the Livestock and Veterinary Services Division at the Ministry of Agriculture and Forestry relies on the existing Integrated Animal Disease Surveillance and Reporting (IADSR) system to collect data on antimicrobial use. Data were collected on antimicrobial use weekly from the community level and submitted to the national level utilizing the IADSR [21]. An operational research study from Sierra Leone that assessed this system for the period 2016–2019 described very poor district reporting rates, complete discordance in data between district and sub-district level treatment forms, and workforce shortage [22]. Based on this report, corrective steps in terms of staff capacity-building were conducted. In this context, we undertook another operational research study to assess antimicrobial use in the treatment of livestock from 15 districts in Sierra Leone from March to October 2021. The three specific objectives were to (1) determine the proportion of weekly reports submitted by the district on livestock health and identify gaps in terms of quality of data gathered, (2) determine the number and proportion of livestock that were found to be sick and received antimicrobials and/or anthelmintics, and (3) assess the extent of usage of OIE antimicrobial agents of veterinary importance, WHO critically important antimicrobials in the treatment of diseases among livestock, and other standard antimicrobial use classifications. This study was considered essential to maintaining a continuum in assessing the improvements in the surveillance system and the status of antimicrobial use among livestock.

## 2. Methods

### 2.1. Study Design

This is a descriptive study based on analysis of routine data transmitted weekly by District Livestock Officers from all the 15 agricultural districts to the Animal Health Epidemiology Unit (AHEU) of the Ministry of Agriculture and Forestry.

### 2.2. Setting

Sierra Leone lies in the western part of Africa and has a population of approximately eight million [23]. Sierra Leone borders the Republic of Guinea in the east and to the north-west and the Republic of Liberia to the south. The country is divided into five regions and 16 districts (15 agricultural districts). Each district is further subdivided into chiefdoms, and each chiefdom consists of many villages. Agriculture (crops, livestock, forestry, and fisheries) is the main source of livelihood in Sierra Leone and contributes to over 75% of livelihoods and about 47% of the country’s gross domestic product (GDP) [24]. According to a survey conducted by the Food and Agriculture Organization of the United Nations in 2016, the livestock population was estimated at 245,736 cattle, 963,001 sheep, 1,567,789 goats, 125,064 pigs, and 14,721,718 poultry [25]. The livestock production system for ruminants has been described generally as traditional, free-range, or extensive low-input.

The Livestock and Veterinary Services Division is one of the six divisions under the Ministry of Agriculture and Forestry (MAF) to promote animal health and production services. Two units form the division: (i) Veterinary Services Unit, which is in charge of animal health, animal disease surveillance, including AMR; public health, inspection, regulations, and certification, as well as animal welfare; and (ii) Animal Production Unit, which is in charge of animal husbandry, animal traction, and animal production activities. The treatment of disease in livestock is guided by the OIE Terrestrial Animal Health Code, Chapter 6.10, which provides guidelines on the responsible and prudent use of antimicrobials in veterinary medicine [26]. The Livestock and Veterinary Services Division has 15 offices in each agricultural district. The division has one district livestock officer (DLO)/veterinary officer (DVO) at the district level, livestock assistants (LA) at the chiefdom level, and community animal health workers (CAHWs) at the community level. The CAHWs are expected to visit farmers regularly to assess the health of their animals and ascertain the number of susceptible animals depending on the exhibition of signs and symptoms of any sickness in any of the animals. Depending on the presentation, the CAHWs can prescribe treatment themselves or consult the LAs or DLO. All these personnel are involved in detecting and treating sick animals, which includes administering antimicrobials.

The CAHWs convey information to the LAs at the chiefdom level or the DLO, depending on proximity about animal sickness gathered during their visits to farmers using a standard reporting format. The LA compiles the data and transmits it to the DLO every week. The DLO compiles all the data collected by the LAs or CAHWs using a standard epidemiological reporting format and transmits the data electronically to the AHEU by 4:00 p.m. every week. A single week’s report submitted by the DLO contains sub-reports from multiple CAHWs. The reports transmitted by the DLOs through the Integrated Animal Disease Surveillance and Reporting System (IADSR) consist of data on type of diseases, species affected, the total number of animals at risk, means of diagnosis, sex, number of sick animals, number of animal deaths, village and chiefdom of the outbreak/disease event, number of animals treated and/or vaccinated, and any actions undertaken. The district’s standard epidemiological template comprises 16 transboundary animal diseases and zoonoses that were prioritized by the country. The priority diseases include peste des petits ruminants (PPR), rinderpest, hemorrhagic septicemia, black quarter/blackleg, contagious bovine pleuropneumonia (CBPP), African swine fever, trypanosomosis, orf (ecthyma contagiosum), brucellosis, tuberculosis, contagious caprine pleuropneumonia (CCPP), avian influenza, anthrax, Newcastle, rabies, foot and mouth (FMD), Rift Valley fever, viral hemorrhagic fevers (Ebola and Lassa fever), salmonellosis and plague. The raw data from the districts are collated, analyzed, and interpreted at the AHEU. Weekly reports are prepared and disseminated through Emergency Preparedness Response Resilience Group (EPRG) meetings every Wednesday and communicated as animal health bulletin to various stakeholders at the national and district level through various Google Groups.

### 2.3. Classification of Antimicrobials

The OIE has categorized the antimicrobials based on the importance of the antimicrobial agents in veterinary sciences as veterinary critically important antimicrobial agents (VCIAs), veterinary highly important antimicrobial agents (VHIAs), and veterinary important antimicrobial agents (VIAs) [27]. The WHO has also identified the animal antimicrobials according to their importance in human medicine into the following four categories: critically important, highly important, important, not important [28,29].

The European Medicines Agency (EMA) and World Health Organization have provided classifications to guide the rational use of antimicrobials in veterinary and human use, respectively. The EMA classified antimicrobials into A (avoid use in food-producing animals), B (restrict veterinary use as these are of critical importance to humans), C (use cautiously only when there are no drugs in Category D that could be clinically effective), and D (first-line drugs to be used prudently) [30]. The WHO AwaRe classification categorizes antimicrobials into access (first- or second-choice drugs with minimal resistance potential), watch (first- or second-choice drugs with specific indications and prone to resistance), and reserve (last-resort drugs for targeted use in multidrug-resistant infections) [19].

### 2.4. Study Participants, Period, Data Sources, Variables, and Data Collection

The study participants include livestock assessed for potential illness and treatment by the LA and CAHWs, documented in the EpiWeek form and reported in the IADSR system from March to October 2021. The district EpiWeek report, an Excel-based district-level weekly report submitted by each district, was the primary data source. From these EpiWeek reports, we collected information on the number of weekly reports received from each district during the study period. From each report, we gathered data on the number of livestock species found to be sick, the type of illness, and the names of antimicrobials or anthelmintics used to treat the illness.

### 2.5. Data Analysis and Statistics

We used frequencies and proportions to summarize the data for our study. We first counted and assessed the number of EpiWeek reports received from each district compared to the expected number of reports (n = 35) during the study period. After that, we described the number and species of livestock that were found to be diseased, type of disease, and antimicrobials used to treat the disease. Each antimicrobial was then manually categorized according to the WHO, OIE, EMA, and AwaRe systems by two study investigators (FIB and AL) and verified independently by a third investigator (DN). The reports were also checked for quality regarding the completeness of reporting fields, uniformity in reporting categorical variables, and plausibility of values.

## 3. Results

### 3.1. Availability of Weekly Reports and Quality of Reporting

At the national level, 461 (88%) out of an expected 525 weekly reports were available from 15 districts over the 35-week study period. Availability of reports was highest from Bombali district (100%) and lowest from Tonkolili district, where reports were available for 26 weeks (74%) (Table 1). The 461 reports which were analyzed for this study comprised 1950 sub-reports. Of these, 1920 reports reported a disease occurrence. Availability of weekly reports by region is presented in Appendix A.

Data quality issues such as those listed in Box 1 were identified in 152 sub-reports which formed 20% (95/461) of all the weekly reports extracted at the district level. In addition, data on certain parameters required for monitoring were not available in the current format or were recorded in a nonuniform manner and could not be analyzed.

Box 1Data quality issues and shortfalls in current reporting formats used to submit Animal Health Weekly Reporting forms by district-level officers in Sierra Leone between March–October 2021
Data quality issues found in 152 out of 1950 sub-reports
o Discrepancies between numbers of susceptible vs. sick vs. treated (14 sub-reports).o Antimicrobial/anthelmintic use was reported, but the list of drugs prescribed did not include any antimicrobial/anthelmintic (97 sub-reports).o Treatment details provided include antimicrobials/anthelmintic drugs but the number treated with antimicrobials/anthelmintics reported as “0” (27 sub-reports).o Missing data in sub-reports
▪Diagnosis missing, though treatment details and numbers of sick animals are provided (22 sub-reports).▪Name of affected species missing (22 sub-reports).
o Humans reported as affected species (11 sub-reports) in cases of dog and monkey bites, with no information on the treatment offered to affected animal species.
Limitations in the design of reporting fields
o No uniform categorizations for locations, diseases, species, and treatment (use of free text fields). Specific name of antimicrobial was not mentioned in 253 sub-reports, and therefore antimicrobial use classification categorization was not possible (253 sub-reports).o Unable to ascertain if the disease reports are from a single farm or multiple farms.o Anthelmintics are also reported under usage of antimicrobials.
Parameters missing in the current format
o Timeliness of reporting. o Number of CAHWs reporting per week.o Level of diagnostic certainty.o Duration of treatment and route of administration of drugs.o Data on follow-up of cases.o Clear disaggregation for sub-reports.



### 3.2. Number of Animals under Surveillance (Susceptible vs. Sick vs. Treated)

Over 35 weeks, 45,267 animals were surveyed by CAHWs, of which 11,883 (26%) were found to have signs or symptoms suggestive of a disease. One-third of all livestock reported to have any kind of disease received anthelmintic drugs, while one-fourth received antimicrobials (Table 2). Almost half of the livestock identified as “sick” were initiated on antimicrobials and/or anthelmintics in species such as cattle, goats, and sheep (Table 2).

### 3.3. Use of Antimicrobials and Anthelmintics

The anthelmintics used included ivermectin, benzimidazoles (albendazole and mebendazole), piperazine, praziquantel, and pyrantel pamoate. The antimicrobials used included tetracyclines (oxytetracycline), macrolides (tylosin), diaminopyrimidines (trimethoprim-sulfamethoxazole), beta-lactams (benzylpenicillin), aminoglycosides (gentamicin), quinolones (ciprofloxacin, levofloxacin), nitroimidazoles (metronidazole), and nozomil (diminazene diaceturate). The name of the antimicrobial was not mentioned in 253 sub-reports, and instead, terms such as antibiotics or antimicrobials were mentioned; hence, antimicrobial use classification could not be carried out. These sub-reports included 967 animals initiated on antimicrobials. The categorization of the antimicrobials is provided in Appendix A.

Out of 3020 animals that were reported to have been initiated on antimicrobial treatment, 77% had been treated with a drug classified as critically important for veterinary use by the OIE, and 60% had been treated with a drug classified as highly important for human use by the WHO (Table 3). Antimicrobials belonging to the Class D of OIE (prudence) or watch group of AwaRe were used in 60% of animals (Appendix A). Oxytetracyclines were the most commonly prescribed antimicrobial used in 56% of the animals, followed by tylosin in 16% of animals treated with antimicrobials (Appendix A).

The drugs prescribed for commonly encountered animal diseases are listed in Table 4. All ailments except rabies and peste des petits ruminants (PPR) require treatment with antimicrobials as per the OIE Terrestrial Animal Health Code Volume 2 [31]. Anthelmintics are indicated in most diseases.

## 4. Discussion

This is a follow-up study undertaken by the Livestock and Veterinary Services Division of Sierra Leone after initiating measures to rectify gaps in the surveillance on animal diseases and antimicrobial use identified previously [22]. This study found that 88% of the expected weekly reports from the 15 districts were received at the national level. This finding has been a significant improvement since the previous operational research study, where weekly reports were available only from three districts, accounting for less than 1% of expected forms [22]. The greatly enhanced number of weekly reports has enabled us to describe the antimicrobial use under routine programmatic conditions—information essential to promote and monitor the rational use of antimicrobials in livestock.

The major strength of this study is the use of nationwide data from the routine surveillance system from the Livestock and Veterinary Services Division of Sierra Leone. Therefore, the study reflects data from all parts of the country without selection bias. The study’s major limitation is that the data were sourced from the routine surveillance system, and data from such sources are known to contain errors. We have not measured the magnitude and direction of these errors to account for them in our interpretation of the results. As there were no benchmarks or targets to be achieved at the field level, and as there were no incentives/disincentives that would have led to deliberate misreporting, we believe that the errors in data in our study are minimal, unintentional, and random. Hence, we strongly believe that the findings are valid and have the following implications on policy and practice for Sierra Leone.

First, though 88% of the weekly reports were received from the district, there are some concerns about the completeness and quality of data. For example, it was not clear whether the weekly district reports incorporated the activities/observations of all CAHWs in their respective districts during the corresponding week and whether these reports were submitted on time. In addition, there was also a lack of uniformity in the terminology used to report the animal disease and the drugs used in their treatment. Addressing these major issues by standardizing the reporting formats, reporting mechanisms, and terminology will greatly strengthen the existing surveillance system. Consultations among stakeholders including policymakers, subject experts, program officials, and field staff involved in IADSR could enable development of standardized reporting mechanisms capable of generating actionable data. Once operationalized, these data should be reviewed periodically to assess performance of the surveillance system and inform decision-making.

Second, this is the first study from Sierra Leone that provides nationally representative information on the prevalence of diseases among livestock, and the data indicate that one-fourth of the livestock assessed by CAHWs during the study period had an infectious disease. However, all diseases were diagnosed by signs and symptoms, and none were laboratory-confirmed. Therefore, the amount of misclassification between sick and healthy livestock is uncertain. Anecdotal evidence indicates constraints with the availability and access to laboratory services to confirm diagnoses. This area needs to be addressed to obtain more accurate information on livestock diseases in the country. Due to the lack of previous data to assess trends in livestock diseases, this study can act as the baseline for monitoring the future trends in livestock diseases in the country.

Third, antimicrobials were used in one-fourth and anthelmintics were used in one-third of the sick animals. It was unclear what proportion of the sick animals remained untreated and why. Available data demonstrated a wide variation in the use of drugs. In most of the sick animals that were treated, the drugs used for the treatment were found to be appropriate, but, for a few diseases, the drugs were inappropriate (e.g., use of anthelmintics and antimicrobials in dogs suspected of rabies and in goats suspected of PPR disease). These indicate deficiencies in the quality of veterinary care and may be improved by training, supportive supervision, and monitoring. Apart from this, the current recording and reporting system is not designed to capture information on the dosage, duration, withdrawal period, and outcomes of the treatment of sick animals under routine programmatic conditions. Therefore, there is a lack of data on the effectiveness of the treatment administered. A veterinary-clinic-based study in Ghana showed a lack of documentation of these parameters in animal treatment records [32]. These parameters will be crucial for any future surveillance of AMR or anthelminthic resistance. If feasible, obtaining information on these parameters can be built into the existing surveillance systems; if not, this is an area for future research.

Fourth, nearly three-fourths of the livestock received antimicrobials belonging to the “veterinary critically important” group under the OIE classification of antimicrobials for veterinary use. As in this study, few other studies from Africa have reported that tetracyclines are the most commonly used drugs in livestock diseases [20,33,34,35]. These drugs are essential for treating certain diseases in livestock, and these drugs may not have substitutes if resistance emerges. This calls for creating systems for enhanced supervision and monitoring of these antimicrobials. In Sierra Leone, the 2020 Animal Health Bill Part 13 specifies which professionals can prescribe veterinary products, including antimicrobials [36]. However, these guidelines are generic (i.e., they do not specify names of the antimicrobials to be used), and the choice of the antimicrobials used is left to the judgement of the CAHWs or LAs at the field level. This may lead to misuse of certain antimicrobials under routine settings, leading to AMR development. Creating more specific guidelines will help the livestock and veterinary services division to regulate, support, and monitor the use of each antimicrobial.

Fifth, most of the antimicrobials used in the country to treat infectious diseases among livestock also belong to the WHO “critically important” or “highly important” group in human medicine and belong to the “watch” category of WHO AwaRe classification [19,29]. Therefore, there is a real threat of augmenting AMR due to the injudicious use of these antimicrobials among animals and also in humans. Since these are known to be at relatively high risk of selection for bacterial resistance, their use must be monitored closely. Considering that these antimicrobials may be the appropriate treatment for some animal ailments, there is an urgent need to establish or strengthen existing AMR surveillance mechanisms among animals and humans. A similar case can be made for anthelmintics. Apart from treating diagnosed helminthic diseases, they are also prescribed precautionary measures during farm visits. Though there is limited information on the risk of resistance to anthelmintics in animals and its implications for humans, it is prudent to initiate monitoring of the use of these drugs in animals [14,37].

Lastly, and most importantly, this study, along with the previous study [22], demonstrates the role of operational research in identifying problems in antimicrobial use and assessing the impact of the corrective measures to address the issues identified in the initial study. To our knowledge, there are very few such follow-up study examples in published literature, and this study should be considered as a role model for similar follow-up studies in other parts of the world.

## 5. Conclusions

This study showed that nearly 88% of the expected district-level weekly reports on livestock diseases and antimicrobial use were received nationwide. Roughly one-fourth of the animals assessed had an infectious disease, and nearly one-fourth of them received an antimicrobial drug. Most of the antimicrobials belonged to the “veterinary critically important” group under the OIE classification of antimicrobials for veterinary use. These antimicrobials also belong to the WHO “critically important” or “highly important” group in human medicine. The study has identified areas for improving the existing surveillance system on antimicrobial use among livestock. In addition, it has highlighted the need for enhancing the AMR stewardship program in both animals and humans.

## Figures and Tables

**Table 1 ijerph-19-05294-t001:** The number of Animal Health Weekly Reporting forms expected and available per district in Sierra Leone, March–October 2021.

District	Number of CAHWs ^1^	Number of Weekly Reports Received Out of Expected ^2^ n (%)	
All districts	151	461/525	(88)
Bo	15	33/35	(97)
Bombali	20	35/35	(100)
Bonthe	6	29/35	(83)
Falaba	7	34/35	(97)
Kambia	6	31/35	(89)
Kailahun	13	31/35	(89)
Kerene	8	30/35	(86)
Kenema	15	28/35	(80)
Kono	14	32/35	(91)
Koinadugu	15	30/35	(86)
Moyamba	3	31/35	(89)
Port Loko	9	29/35	(83)
Pujehun	10	31/35	(89)
Tonkolili	8	26/35	(74)
Western area	2	31/35	(89)

^1^ Community animal health worker. ^2^ Expected number of forms = number of weeks in the study period (EpiWeek 9 to 43).

**Table 2 ijerph-19-05294-t002:** Number of livestock treated with antimicrobials or anthelmintic drugs used by animal species in Sierra Leone, March–October 2021 ^1^.

Livestock Species (as Reported)	Susceptible Livestock	Sick Animals		Livestock Treated with Antimicrobials		Livestock Treated with Anthelmintics	
	n	n	(%) ^2^	n	(%) ^3^	n	(%) ^3^
Cattle	1175	362	(30.8)	168	(46.4)	189	(52.2)
Dogs	711	117	(16.5)	43	(36.8)	38	(32.5)
Donkeys	7	4	(57.1)	1	(25.0)	0	(0.0)
Fowl	6569	914	(13.9)	3	(0.3)	60	(6.6)
Goat	22,198	6229	(28.1)	1576	(25.3)	2036	(32.7)
Goats and Sheep ^4^	1409	835	(59.3)	402	(48.1)	399	(47.8)
Horse	40	16	(40.0)	15	(93.8)	16	(100.0)
Pig	1930	574	(29.7)	105	(18.3)	220	(38.3)
Rabbit	52	30	(57.7)	4	(13.3)	23	(76.7)
Sheep	10,775	2691	(25.0)	692	(25.7)	991	(36.8)
Not recorded	401	111	(27.6)	11	(2.7)	68	(16.9)
Total	45,267	11,883	(26.2)	3020	(25.4)	4040	(33.9)

^1^ Reports with discrepancies between numbers of susceptible vs. sick vs. treated were excluded from analysis. ^2^ Percentage of sick out of susceptible animals. ^3^ Percentage of livestock initiated on antimicrobial or anthelmintic treatment out of sick animals. ^4^ Some reports mentioned species as “goats and sheep”, and disaggregated species-wise information was not available, hence retained as a separate category.

**Table 3 ijerph-19-05294-t003:** Antimicrobial use based on OIE list of antimicrobial agents of veterinary importance and World Health Organization list of critically important antimicrobials for human medicine in Sierra Leone, March–October 2021 ^1^.

Livestock Species (as Reported)	Number of Livestock Treated with Antimicrobials	OIE Classification of Antimicrobials for Veterinary Use ^3^		WHO Critically Important Antimicrobials for Human Medicine Categories					
		Veterinary Critically Important		Critically Important		Highly Important		Important	
	N	n	(%) ^2^	n	(%) ^2^	n	(%) ^2^	n	(%) ^2^
Cattle	168	145	(86.3)	30	(17.9)	115	(68.5)	0	(0.0)
Dogs	43	37	(86.0)	17	(39.5)	20	(46.5)	0	(0.0)
Donkeys	1	0	(0.0)	0	(0.0)	0	(0.0)	0	(0.0)
Fowl	3	3	(100.0)	0	(0.0)	3	(100.0)	0	(0.0)
Goat	1576	1070	(67.9)	290	(18.4)	781	(49.6)	4	(0.3)
Goats and Sheep ^4^	402	402	(100.0)	31	(7.7)	371	(92.3)	0	(0.0)
Horse	15	15	(100.0)	0	(0.0)	15	(100.0)	0	(0.0)
Pig	105	89	(84.8)	8	(7.6)	81	(77.1)	0	(0.0)
Rabbit	4	4	(100.0)	0	(0.0)	4	(100.0)	0	(0.0)
Sheep	692	557	(80.5)	142	(20.5)	415	(60.0)	3	(0.4)
Not recorded	11	3	(27.3)	1	(0.9)	2	(18.8)	0	(0.0)
Total	3020	2325	(76.9)	519	(17.1)	1807	(59.8)	7	(0.2)

^1^ Reports with discrepancies between numbers of susceptible vs. sick vs. treated; multiple antimicrobials belonging to different categories may be used for the same condition. ^2^ Percentages calculated out of all livestock initiated on antimicrobial treatment. ^3^ Since all classifiable antimicrobials in the data belonged to the OIE critically important category, the remaining two OIE categories are not shown in the table. ^4^ Some reports mentioned species as “goats and sheep”, and disaggregated species-wise information was not available, hence retained as a separate category.

**Table 4 ijerph-19-05294-t004:** Prescription practices by species for common illness in Sierra Leone, March–October 2021.

Livestock Species	Condition	Commonly Used Drugs	Treatment Recommended as Per Guidelines ^1^
Cattle	Infectious pododermatitis	Albendazole, Penicillin, Ivermectin, Tylosine and Sulphamethoxazole	Antibiotic
	Mange	Oxytetracycline, Ivermectin	Antibiotic + anti-parasitic (injectable)
	Tick infestation	Oxytetracycline, Tylosine, Ivermectin	Antibiotic + anti parasitic (injectable) + bath solution (tik-stop)
	Worm infestation	Mebendazole, Oxytetracycline, Ivermectin	Anthelminthic + antibiotic
Dogs	Suspected rabies	Albendazole, Oxytetracycline, Ivermectin	Anti-rabies vaccine (if dog found not showing signs and symptoms after quarantine)
	Mange	Oxytetracycline, Ivermectin	Antibiotic + anti-parasitic (injectable)
Fowl	Newcastle disease	Piperazine, Oxytetracycline, Ivermectin	Newcastle Vaccine + multivitamin + antibiotic
Goat	Mange	Oxytetracycline, Ivermectin	Antibiotic + anti-parasitic (injectable)
	Peste des petits ruminants (PPR)	Albendazole, Oxytertracycline, Ivermectin	PPR vaccine
	Infectious pododermatitis	Oxytetracycline, Ivermectin, Gentamicin, Kenflox (Ofloxacin + Orindazole), Albendazole	Antibiotic
Horse	Mange	Oxytetracycline, Ivermectin	Antibiotic + anti-parasitic (injectable)
Pigs	Mange	Oxytetracycline, Ivermectin	Antibiotic + anti-parasitic (injectable)
Sheep	Mange	Oxytetracycline, Ivermectin	Antibiotic + anti-parasitic (injectable)
	Infectious pododermatitis	Oxytetracycline, Ivermectin Sulfamethoxazole-Trimethoprim, Kenflox (Ofloxacin + Orindazole), Albendazole, Lemoxine	Antibiotic
	Peste des petits ruminants (PPR)	Albendazole, Oxytetracycline, Ivermectin	PPR vaccine
Rabbit	Mange	Oxytetracycline, Ivermectin	Antibiotic + anti-parasitic (injectable)

^1^ OIE Terrestrial Animal Health Code Volume 2.

## Data Availability

Requests to access these data should be sent to the corresponding author.

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
