# Peer review of "An Update on the Surveillance of Livestock Diseases and Antimicrobial Use in Sierra Leone in 2021—An Operational Research Study"

_ijerph, 2022, doi:10.3390/ijerph19095294_

Round 1
Reviewer 1 Report
In my opinion your paper entitled “ An update on the surveillance of livestock diseases and antimicrobial use in Sierra Leone in 2021—An operational research study " is an original , well conceptualized work with interesting results. I think that the scientific contribution of this research is very significant in the field of livestock diseases and AMR monitoring. Thus, the manuscript should be accepted for publication.
The study is well designed and that the data obtained are important for further work and monitoring of livestock in Sierra Leone. The authors also explained the advantages and disadvantages of the study, which I see as an space for further improvement of data collection methods and improvement of services for providing therapy to sick animals. Given that the authors themselves pointed out the shortcomings of their research, the only thing I would add is that they could write what their plans are for the future in order to overcome the weaknesses of the study that they themselves stated.
Author Response
Point 1: In my opinion your paper entitled “ An update on the surveillance of livestock diseases and antimicrobial use in Sierra Leone in 2021—An operational research study " is an original , well conceptualized work with interesting results. I think that the scientific contribution of this research is very significant in the field of livestock diseases and AMR monitoring. Thus, the manuscript should be accepted for publication.
The study is well designed and that the data obtained are important for further work and monitoring of livestock in Sierra Leone. The authors also explained the advantages and disadvantages of the study, which I see as an space for further improvement of data collection methods and improvement of services for providing therapy to sick animals. Given that the authors themselves pointed out the shortcomings of their research, the only thing I would add is that they could write what their plans are for the future in order to overcome the weaknesses of the study that they themselves stated..
Response 1: We thank the reviewer for their observation. As mentioned in the manuscript (Box 1), the existing Integrated Animal Disease Surveillance and Reporting (IADSR) database from which the data was extracted for analysis had some shortfalls due to which a comprehensive analysis of performance could not be carried out. As a corrective measure, we had suggested standardizing the reporting formats, reporting mechanisms, and terminology used for reporting (Line 293-295). Consultations among stakeholders including policy makers, subject experts, programme officials and field staff involved in IADSR could enable development of standardized re-porting mechanisms capable of generating actionable data. Once operationalized, this data should be reviewed periodically to assess performance of the surveillance system and informed decision making. This has been added in the revised manuscript (Line 295-300).
Reviewer 2 Report
This paper explores the reporting of livestock disease and treatment in Sierra Leone. The paper is of wider interest in providing information on possible incidence of disease and use of antibiotics, but particularly in identifying issues with diagnosis, treatment and reporting, which may well also be occurring in other regions. A key strength of the paper is the size of the data set used. The methods are generally well described, and the qualitative presentation is appropriate. Deficiencies in reporting (eg names of disease - what is included as 'footrot'?) which are apparent in the results have been highlighted in the discussion as procedural issues which should be improved, overcoming concerns that the method lacked adequate rigour. The paper could be improved with some minor revisions to writing style.
Minor comments:
Abstract - first sentence is long and complicated - please revise.
Last sentence in the abstract - change in font size?
There are some places where the use of the same word occurs in the same sentence. I suggest re-wording to avoid repetition eg page 2, 2nd last paragraph 'The strategic goals of the national strategic plan....' and Methods section 2.1 ' Study...study'.
p6 Box 1 - Not sure that 'Box' is the appropriate descriptor.
Table 4 - Should this table include scientific names of diseases eg what is meant by footrot? If diagnosis was not accurate by veterinarians and the disease listed could actually be several conditions, this needs to be made clear in the methods/table.
Table S4 - table formatting incorrect.
Author Response
Point 1: This paper explores the reporting of livestock disease and treatment in Sierra Leone. The paper is of wider interest in providing information on possible incidence of disease and use of antibiotics, but particularly in identifying issues with diagnosis, treatment and reporting, which may well also be occurring in other regions. A key strength of the paper is the size of the data set used. The methods are generally well described, and the qualitative presentation is appropriate. Deficiencies in reporting (eg names of disease - what is included as 'footrot'?) which are apparent in the results have been highlighted in the discussion as procedural issues which should be improved, overcoming concerns that the method lacked adequate rigour. The paper could be improved with some minor revisions to writing style.
Response 1: We thank the reviewer for these inputs. We have attempted to make corrections in the writing style in the revised manuscript and hope these are found to be satisfactory.
Point 2: Abstract - first sentence is long and complicated - please revise.
Response 2: The sentence has been simplified (Line 28-30)
Point 3: Last sentence in the abstract - change in font size?
Response 3: The last sentence in the abstract has been omitted from the latest revision of the manuscript, as suggested by one of the reviewers.
Point 4: There are some places where the use of the same word occurs in the same sentence. I suggest re-wording to avoid repetition eg page 2, 2nd last paragraph 'The strategic goals of the national strategic plan....' and Methods section 2.1 ' Study...study'.
Response 4: Corrections have been made in Line 84 and Line 108
Point 5: p6 Box 1 - Not sure that 'Box' is the appropriate descriptor.
Response 5: Since “Box” is a generally accepted form of presentation for such data/information (as shown in Box 1) in scientific journals, we request it may be retained in the manuscript.
Point 6: Table 4 - Should this table include scientific names of diseases eg what is meant by footrot? If diagnosis was not accurate by veterinarians and the disease listed could actually be several conditions, this needs to be made clear in the methods/table.
Response 6: The term “Foot Rot” has been replaced with the appropriate scientific name, i.e, “Infectious pododermatitis” in the manuscript. The rest of the diseases have been listed as reported in the surveillance system.
Point 7: Table S4 - table formatting incorrect.
Response 7: Table formatting has been corrected and uploaded as a revised supplementary file.
Reviewer 3 Report
This study uses nationwide data but for a limited period of time and not for the entire year 2021 to inform us on the prevalence of diseases among livestock. None of the diseases reported were laboratory confirmed but diagnosed by signs and symptoms. The information is not accurate on livestock diseases. In addition, there is no specific information on resistance to antibiotics and anthelminths as well as there is a lack of data on the effectiveness of the treatment.
Author Response
Point 1: This study uses nationwide data but for a limited period of time and not for the entire year 2021 to inform us on the prevalence of diseases among livestock. None of the diseases reported were laboratory confirmed but diagnosed by signs and symptoms. The information is not accurate on livestock diseases. In addition, there is no specific information on resistance to antibiotics and anthelminths as well as there is a lack of data on the effectiveness of the treatment.
Response 1: We thank the reviewer for their comment.
We agree that the estimates of disease prevalence may not be reliable and the same has been explicitly mentioned in Lines 301-310. Similarly, due to limitations of the available data described in the Results, we have refrained from making any comment on effectiveness of treatment or resistance patterns. This aspect has also been highlighted and we have made recommendations for strengthening of the surveillance system to allow for these analysis in the discussion section (line numbers 338-358).
We would like to reiterate that this study was designed as a follow up operational research activity based on analysis of data from an existing programmatic veterinary disease surveillance system. If the lacunae identified through this effort are addressed, we strongly feel that this will lead to reporting of better data which can answer research questions on prevalence, effectiveness and resistance patterns.
Reviewer 4 Report
The manuscript of Bangura et al. 2022 is an update on the surveillance of livestock diseases and antimicrobial use in Sierra Leone. The work shows the improvement in the animal health surveillance system in Sierra Leone and highlights the need for enhanced antimicrobial stewardship.
Comments:
1) The last two phrases of the abstract are similar, please revise.
2) The manuscript have several abbreviations, please revise if their complete meaning is described on the text. For example, ORF (pag. 4) is missing the meaning of this abbreviation.
3) Table 1: The 2 columns with number (n) and the one with percentage can be merged. Or in the upper line of these columns the "n" and "%" should be aligned with the respective column. For easier visualization.
4) The references 11, 13, 14, 16, 19, 21, 23, 24, 26, 27, 29, 31, 37 are incomplete. Please confirm.
Author Response
Point 1: The last two phrases of the abstract are similar, please revise.
Response 1: We thank the reviewer for bringing this to our notice. The last sentence of the abstract has been removed from the revised manuscript.
Point 2: The manuscript have several abbreviations, please revise if their complete meaning is described on the text. For example, ORF (pag. 4) is missing the meaning of this abbreviation.
Response 2: Orf (ecthyma contagiosum) is a zoonotic viral skin infection and was mistakenly capitalised as ORF. This has now been corrected (Line 158). We have also reviewed the manuscript again to ensure that abbreviations are expanded at the first instance of their use in the text, as per journal guidelines.
Point 3: Table 1: The 2 columns with number (n) and the one with percentage can be merged. Or in the upper line of these columns the "n" and "%" should be aligned with the respective column. For easier visualization.
Response 3: Table 1 has been re-formatted for better visualisation.
Point 4: The references 11, 13, 14, 16, 19, 21, 23, 24, 26, 27, 29, 31, 37 are incomplete. Please confirm.
Response 4: References have been corrected as follows and have been updated in the revised manuscript.
- World Health Organisation. Global action plan on antimicrobial resistance [Internet]. 2017 [cited 2022 Mar 28]. Available from: https://www.who.int/publications-detail-redirect/9789241509763
- Food and Agriculture Organisation (FAO). The FAO Action Plan on Antimicrobial Resistance 2016-2020 |Policy Support and Governance| Food and Agriculture Organization of the United Nations [Internet]. 2016 [cited 2022 Mar 28]. Available from: https://www.fao.org/policy-support/tools-and-publications/resources-details/en/c/459933/
- World Organisation for Animal Health. Responsible and prudent use of anthelmintic chemicals to help control anthelmintic resistance in grazing livestock species [Internet]. 2021 [cited 2022 Mar 28]. Available from: https://www.oie.int/en/document/anthelmintics-grazing-livestock-2021/
- Grace D. Review of evidence on antimicrobial resistance and animal agriculture in developing countries [Internet]. Evidence on Demand; 2015 Jun [cited 2022 Mar 28]. Available from: https://cgspace.cgiar.org/handle/10568/67092
- World Health Organization. The 2019 WHO AWaRe classification of antibiotics for evaluation and monitoring of use [Internet]. World Health Organization; 2019 [cited 2022 Mar 28]. Report No.: WHO/EMP/IAU/2019.11. Available from: https://apps.who.int/iris/handle/10665/327957.
- Government of Sierra Leone. Sierra Leone: National Strategic Plan for Combating Antimicrobial Resistance [Internet]. 2017 [cited 2022 Mar 28]. Available from: https://www.who.int/publications/m/item/sierra-leone-national-strategic-plan-for-combating-antimicrobial-resistance
- MOHS. Housing and population census. Demographic and Health Survey 2015. Freetown, Sierra Leone. Available: https://www.statistics.sl/index.php/census/census-2015.html (cited 2022 Mar 28).
- Sesay AR. Review of the livestock/meat and milk value chains and policy influencing them in Sierra Leone [Internet]. Rome, Italy: FAO; 2016 [cited 2022 Mar 28]. 66 p. Available from: https://www.fao.org/documents/card/en/c/87ed4679-429f-4d1f-958a-6a0ed5ce7a63/.
- OIE - World Organisation for Animal Health. Chapter 6.8. Harmonisation of national antimicrobial resistance surveillance and monitoring programmes. In: OIE standards, gidelines and resolution on antimicrobial resistance and the use of antimicrobial agents. 2nd edition. Paris: OIE; 2020.
- OIE - World Organisation for Animal Health. List of antimicrobial agents of veterinary importance [Internet]. 2019 [cited 2022 Mar 28]. Available from: https://www.oie.int/en/document/a_oie_list_antimicrobials_june2019/
- World Health Organisation. Critically important antimicrobials for human medicine : 6th revision [Internet]. [cited 2022 Mar 28]. Available from: https://www.who.int/publications-detail-redirect/9789241515528.
- OIE - World Organisation for Animal Health. Volume 2. In: Terrestrial Code Online Access [Internet]. 2021 [cited 2022 Mar 28]. Available from: https://www.oie.int/en/what-we-do/standards/codes-and-manuals/terrestrial-code-online-access/.
- Government of Sierra Leone. Animal Health Bill (Draft) 2020. Sierra Leone; 2020.
Round 2
Reviewer 3 Report
I believe that the key points of my objections remain unresolved. I do not consider the study to be complete for presentation in a journal of high scientific impact such as this one
Author Response
Comment: I believe that the key points of my objections remain unresolved. I do not consider the study to be complete for presentation in a journal of high scientific impact such as this one.
Response: We appreciate the reviewer’s concern. We agree that all the metrics (accurate data, information on laboratory confirmed diagnosis, drug sensitivity patterns, treatment effectiveness) as indicated by the reviewer are important and needed. But unfortunately such information is not available under routine programatic conditions (Integrated Animal Disease Surveillance and Reporting) in Sierra Leone and this is what we wanted to highlight in this operational research study. We feel that highlighting this gap in this esteemed journal will draw attention of policy makers (both national and international) to this issue. As indicated in our manuscript this is a follow-up study and though there has been considerable improvement, the situation is far from ideal. We feel that we have brought out this issue clearly in our manuscript and we welcome any suggestions for strengthening the messages related to the urgent need for further improvement.